# Development of Nontoxic Biodegradable Polyurethanes Based on Polyhydroxyalkanoate and L-lysine Diisocyanate with Improved Mechanical Properties as New Elastomers Scaffolds

**DOI:** 10.3390/polym11121927

**Published:** 2019-11-22

**Authors:** Cai Wang, Jiapeng Xie, Xuan Xiao, Shaojun Chen, Yiping Wang

**Affiliations:** 1Guangdong Research Center for Interfacial Engineering of Functional Materials, Shenzhen Key Laboratory of Polymer Science and Technology, Shenzhen Key Laboratory of Special Functional Materials, Nanshan District Key Lab for Biopolymers and Safety Evaluation, College of Materials Science and Engineering, Shenzhen University, Shenzhen 518060, China; wangcai@szu.edu.cn (C.W.); xiejiapeng2018@email.szu.edu.cn (J.X.); 2170344429@email.szu.edu.cn (X.X.); 2Key Laboratory of Optoelectronic Devices and Systems of Ministry of Education and Guangdong Province, College of Optoelectronic Engineering, Shenzhen University, Shenzhen 518060, China

**Keywords:** L-lysine diisocyanate, biodegradable, polyurethanes, non-toxic, mechanical performance

## Abstract

A nontoxic and biodegradable polyurethane was prepared, characterized, and evaluated for biomedical applications. Stretchable, biodegradable, and biocompatible polyurethanes (LPH) based on L-lysine diisocyanate (LDI) with poly(ethylene glycol) (PEG) and polyhydroxyalkanoates(PHA) of different molar ratios were synthesized. The chemical and physical characteristics of the LPH films are tunable, enabling the design of mechanically performance, hydrophilic, and biodegradable behavior. The LPH films have a Young’s modulus, tensile strength, and elongation at break in the range of 3.07–25.61 MPa, 1.01–9.49 MPa, and 102–998%, respectively. The LPH films demonstrate different responses to a change of temperature from 4 to 37 °C, with the swelling ratio for the same sample at equilibrium varying from 184% to 151%. In vitro degradation tests show the same LPH film has completely different degradation morphologies in pH of 3, 7.4, and 11 phosphate buffered solution (PBS). In vitro cell tests show feasibility that some of the LPH films are suitable for culturing rat bone marrow stem cells (rBMSCs), for future soft-tissue regeneration. The results demonstrate the feasibility of the LPH scaffolds for many biomedical applications.

## 1. Introduction

An ideal tissue-engineering material needs to have the mechanical properties similar to the natural tissue material. Compared with the native tissue regeneration process, the material should be biocompatible and biodegradable [1,2]. Synthesis of a novel biodegradable and biocompatible polyurethane material is important for biomedical engineering applications. Different kinds of polyurethanes are widely used in biomedical applications due to their biostability, biodegradability, and tunableness of the chain-segment structure from an extensive range of available raw materials, such as long chain diols, diisocyanates, and short chain diols (chain extenders) [3,4,5]. It is well-known that conventional polyurethane is composed of soft segments and hard segments. Long chain diols form the soft segments, meanwhile, alternating diisocyanates and short chain diols form the hard segments in the typically phase-separated structure of polyurethane copolymers [6,7,8,9,10]. A great variety of raw materials are available for synthesis polyurethane, which provides possibilities to tailor the chain-segment structure to obtain polymers with different physical and chemical properties, subsequently different mechanical and biodegradable properties [11,12].

Recently, several biodegradable plastic polymers, such as poly(glycolic acid) (PGA) [13], polylactic acid (PLA) [14,15], polycaprolactone (PCL) [16,17], polyhydroxyalkanoates (PHA) [18], and their block copolymers [19,20,21], were developed for biomedical applications. These biodegradable polymers are aliphatic polyesters that can be degraded by hydrolysis, and their degradation products are nontoxic [22]. Compared with the other biodegradable plastic polymers, PCL is slowly degraded and has good flexibility [23]. Notably, PHA is more hydrophilic and has a better biocompatible and faster degradation rate than PCL [24]. PHA is the only fully biosynthetic natural polymeric material which can be completely synthesized by the bacteria [25]. PHA has absorbability, biocompatibility, biodegradability, and thermoplasticity, which make it promising in the field of biomaterials [26]. Unfortunately, conventional PHA polymer material has a high temperature for thermal curing process, which hampered its applications [27,28]. Therefore, the PHA polymer material needs to be modified. The modified PHA-based polymers have low-temperature crosslinking and desirable wettability, while preserving or improving their unique elasticity. It is well-known that the hydrophilicity of the biomaterials can directly determine their degradation and cytocompatibility properties [29]. Furthermore, the hydrophilic segment is usually synthesized by polyethylene glycol (PEG), which is a well-established nonimmunogenic and nontoxic water-soluble polymer. The hydrophilic of biomaterials could prevent the surface of biomaterial interactions with proteins [30,31,32]. 

In this study, a nontoxic biodegradable polyurethane was synthesized by PEG, PHA of different molar ratio, and LDI. Synthetic biocompatible polyurethane select LDI as raw materials, which is due to its nontoxic degradation products [33]. We hypothesized that the synthesis of the LPH is achieved by using the diisocyanate component of LDI, hydrophilic PEG, and biodegradable PHA-diol as the main polyol component. By varying the molar fraction of PHA-diol, the properties of the PHA-based polyurethane could be tailored. The chemical structure and properties of the LPH polymer materials were studied, and the potential biomedical application for LPH was also evaluated. The structure of the LPHs was characterized by spectroscopic and calorimetric measurements. The mechanical properties were analyzed via quasi-static tests, and the swelling behavior was also investigated. In vitro degradation tests of the LPH polymer were carried out in different pH of phosphate buffered solution (PBS). Cytotoxicity was tested by CCK-8 and cell proliferation.

## 2. Materials and Methods

### 2.1. Materials

L-lysine diisocyanate (LDI), Polyethylene glycol (PEG), 1,4-butanediol (BDO), P-toluenesulfonic acid (PTSA), and dibutyl tin dilaurate (DBTDL) were purchased from Aladin, Shanghai, China. PEG was dehydrated at 60 °C for 24 h under vacuum. Polyhydroxybutyrate (PHA, 5.5 × 10^5^ g/mol, 10%4HB) powders were purchased from TianAn Biologic Materials Co., Ltd. (Shandong, China). Organic solvents (ethanol, dimethyl formamide, and chloroform) were used as received. Rat bone marrow stem cells (rBMSCs) were purchased from Shanghai institute of biochemical cells (Shanghai, China). Dulbecco’s modified eagle medium (DMEM) and reagents (fetal bovine serum, penicillin, and mito-tracker green) were purchased from Thermo Fisher Scientific (MA, USA).

### 2.2. Preparation of LPH Films

The superior bio-elastomers (LPH) were synthesized, using a stepwise polymerization process by bridging L-lysine diisocyanate (LDI) with prepolymer. The synthesis scheme of PHA-diol and the synthesis scheme of LPH film are shown in Figure 1. The PHA-diol was obtained by solvent evaporation, according to published methods [34]. Figure 1A shows the PHA-diol powders’ preparation process. The PHA powders were first purified by the recrystallization process. They were dissolved in CHCl_3_ at, 80 °C, by adding the p-toluenesulfonic acid, which reacted with 1,4-butanediol for depolymerization. After 4 h, these powders needed further purified by extraction, precipitation, and filtration. The PHA-diol powders were obtained. It can be seen that the SEM of PHA powder is spherical (d = 20 µm), but the diameter of the PHA-diol spheres was 500 µm. From Figure 1B, the PHA-diols were reacted with prepolymer. LDI and PEG were introduced into DMF solvent, adding 2d DBTDL as catalyst. The prepolymer reaction was at 60 °C for 4 h, by magnetic stirring, and then the PHA-diols was added the reaction solution, at the temperature of 80 °C, for 2 h. At last, the mixture was poured into a Teflon dish, which dried at 60 °C for 48 h, in order to remove the residual solvent. The LPH films were labeled as LPH-0, LPH-2, LPH-4, LPH-6, and LPH-8. The formulation of LPH film is shown in Table 1.

### 2.3. Characterization

Nuclear magnetic resonance (NMR) spectra was obtained using a Bruker (500 MHz). The spectrometer with Chloroform-d or DMSO-d6 as solvent and tetramethylsilane (TMS) as an internal standard. The ATR/FT-IR spectra measurements were carried out on a VERTEX 70 (Bruker, Karlsruhe, Germany). The spectra of LPH films was recorded from 4000 to 500 cm^−1^ in absorbance mode. The chemical structure of the PHA, PHA-diol and LPH were identified by NMR and FT-IR.

The thermal stability of LPH films was investigated by thermo-gravimetric analysis (TGA) on a TA instrument (Q50, TA Instruments, New Castle, PA, USA) under nitrogen atmosphere. The LPH samples in the temperature range from 25 to 500 °C at a heating rate of 10 °C/min. Then TGA data were analyzed in the form of mass loss and the mass loss rate with respect to the temperature. The derivative of TG (DTG) was defined as the rate of change for the degree of conversion. The weight retained for LPH samples was plotted as a function of temperature.

The thermal properties of LPH films were carried out by a differential scanning calorimeter (DSC, Q200, TA Instruments, New Castle, PA, USA). The LPH samples were cooled to −40 °C and then heated to 180 °C, at a heating rate of 10 °C/min, under N_2_ gas. The crystallinity of LPH films was measured by using a wide-angle X-ray diffraction (XRD, Rigaku, Kyoto, Japan) with a scanning rate of 10 °C/min. The diffraction angles were set as 10°–50°. Diffraction patterns were obtained by using a high-resolution diffractometer. Samples were measured in reflection mode.

Morphologies of the LPH films before and after degradation were observed by a scanning electron microscope (SEM, Hitachi S-4800, Japan) at an accelerating voltage of 5.0 kV. The LPH films were prepared by drying in 60 °C oven for 24 h, and the surfaces of the samples before and after degradation by sputtering gold for 50 s.

Tensile testing was performed with CMT4204 (Shanghai, China) at room temperature. The LPH samples were obtained by the dumbbell-shaped mold (50 mm × 8.5 mm × 2 mm; narrow parallel section 16 mm × 4 mm) after solvent evaporation. The tensile properties of the LPH films were evaluated by a SANS system, at room temperature, equipped with a 2 kN load force. The strain rate was 0.01 s^−1^. The tension test for each LPH sample was repeated three times.

### 2.4. Contact Angle

Contact angle measurement was utilized to investigate the hydrophilic performance of the LPH film. The measurement was carried out at room temperature, using 2 μL drops of distilled water at 10 s, following deposition via needle onto the film surface. Contact angles of each LPH film were recorded by a CAM200 (KSV Instruments, Stockholm, Finland). For LPH films, the reported values are an average of three measurements taken on different areas of the same LPH samples.

### 2.5. Swelling Properties

Water absorption experiments were performed by immersing the different LPH films in the deionized water at a selected-temperature (4, 25, and 37 °C) water bath. Until the equilibrium state, the LPH film was removed from the deionized water after 24 h, and excess water on the LPH surface was wiped by filter paper. At last, the LPH film was weighed by an analytical balance. The water absorption of the swollen LPH film was calculated as follows:$$\mathrm{Water}\;\mathrm{absorption}(\%)=\frac{M_{2}\;-\;M_{1}}{M_{1}}\text{×}100$$
where, M_2_ is the absorbing weight of LPH film at different temperature, and M_1_ is the dry weight of the LPH film. Each sample was measured three times.

### 2.6. Cell Viability Test

Cytotoxicity was quantitatively assessed by the CCK-8 method on 1, 3, and 7days after incubation. All the LPH samples were placed in a 24-wells plate and irradiated with ultraviolet light for 60 min. The Cell Counting Kit-8 (CCK-8, Dojindo, Kyushu, Japan) assay assessed the cytotoxicity of LPH samples on 1, 3, and 7 days after rat bone marrow stem cells (rBMSCs) incubation. The BMSCs were cultured on the LPH samples in the 24-well plate, at a density of 1 × 10^5^ cells per well, in low-glucose Dulbecco’s modified Eagle’s medium (DMEM) supplemented with 10% fetal bovine serum (FBS). Only rBMSCs with the same density were cultured in the 24-well plate as the control group. The rBMSCs were kept in a 5% CO_2_ with 100% humid atmosphere incubator at 37 °C. All the LPH samples were transferred into a new 24-well plate containing DMEM medium and CCK-8 reagent at 1, 3, and 7 days, respectively. After incubation at 37 °C, the optical density (OD) values of the supernatant at 450 nm were recorded by using a microplate reader (MULTISKAN MK3, Thermo, Waltham, MA, USA). For each group, all the LPH samples were measured in triplicate.

Cultivation of the rBMSCs was designed to study the possible effects of LPH samples in the cell-proliferation experiments. The rBMSCs were first stained with cell-tracker green. Then, the rBMSCs were cultured on the surface of the LPH samples in a density of 1 × 10^5^ cells/mL. The morphology of the rBMSCs was observed by confocal laser scanning microscope (CLSM) (Leica, TCS SP5Ⅱ, Frankfurt, Germany) on 1, 3, and 7 days, respectively.

### 2.7. Statistical Analysis

All data were presented as mean ± standard deviation (SD), and statistical analysis was performed, using one-way analysis of variance (ANOVA). Statistical significance was defined as *p* < 0.05.

## 3. Results and Discussions

### 3.1. Structural Analysis

The chemical structure of LPH, PHA, and PHA-diol was characterized with ^1^H-NMR. From Figure 2, the ^1^H-NMR spectra of the PHA and PHA-diol were quite similar. However, several new peaks of PHA-diol appeared at 1.62, 1.76, 3.64, and 4.30 ppm. The peak intensity of PHA-diol weakened at 1.2, 2.28, 2.41, 2.53, 4.03, and 5.18 ppm. This may indicate that PHA can be depolymerized and butylene glycol was successfully grafted on PHA [35]. The synthesis PHA-diol reaction process is shown in Figure 1A. Figure 2 depicts the ^1^H-NMR spectra of LPH-8, where the characteristic peaks for PHA-diol, LDI, and PEG can be found. Results showed that PHA-diol was successfully grafted to the end of polyurethane prepolymer, which is consistent with Figure 1B. The LPH-8 in Figure 2 is characteristic of the resonance signal peaks at 7.16 and 5.68 ppm, which is assigned to the protons of amino in the carbamate moiety and methine of PHA, respectively. The presence of protons at 4.01 and 3.31 (–CH_2_CH_2_O–) ppm indicates the presence of PEG. The peaks at 2.93 and 2.91 (–CH_2_CH_2_CH_2_–) ppm are assigned to the methylene protons of the LDI. The peaks at 1.32 (–CH_2_CH_2_CH_2_–) and 1.20 (–CH_3_) ppm are assigned to the methylene protons of the PHA-diol units. The peaks at 2.49 ppm are due to DMSO-d_6_ solvent.

ATR/FT-IR was also used to characterize the structure of the synthesized PHA-based polyurethanes. In Figure 3A, the spectra of PHA and PHA-diol is presented. There is one obvious difference for PHA and PHA-diol at 3343 cm^−1^. The broad peak is the characteristic adsorption of–OH stretching vibrations in the spectra of PHA-diol. For PHA, there is no peak at about 3343 cm^−1^. It means there is no hydroxyl group in PHA. A strong and sharp absorption band at 1724 cm^−1^ in the spectra of PHA and PHA-diol is observed in Figure 3A, which is the–C=O stretching peaks. In Figure 3B, the spectra of the LPH films with different PHA-diol content were quite similar, but there are still some differences between the relative intensities of several bands. FT-IR has been widely used to investigate the hydrogen bonding interaction [36]. The N–H stretching vibration (3200–3500 cm^−1^) and the carbonyl C=O stretching vibration amide I region (1650–1750 cm^−1^) is for polyurethanes’ two principal vibrational regions [37]. Herein, ATR/FT-IR spectra corresponding to the amine region and carbonyl region for polyurethane with different PHA-diol contents are also presented in Figure 3B. A signal characteristic peak for N–H stretching vibrations of the urethane linkage is that the broad band occurs at 3343 cm^−1^, which implies the presence of hydrogen bonds in the LPH polymer backbone. The hydrogen-bonded N–H stretching is formed by the urethane and urea groups [38,39]. The signal peaks at 2919 cm^−1^ (asymmetric stretching) and 2863 cm^−1^ (symmetric stretching) result from CH_2_ in polymer chains. The signal characteristic peak at 1721 cm^−1^ of LPH films, except LPH-0, shows the presence of free carbonyl (C=O) in polymer backbone, but the signal peak at 1708 cm^−1^ is attributed to the hydrogen bond in urethane carbonyl (NH–C=O), which is consistent with literature [40,41]. The peak intensity of the hydrogen bond increases in the LPH films (from LPH-0 to LPH-8). The characteristic peaks at 1622, 1544, and 1539 cm^−1^ are attributed to the bending vibrations of the N–H bond. The signal characteristic peak of the C–N bond appears at 1350 cm^−1^. A signal characteristic peak of C–C occurred at 1452 cm^−1^. The absorption bands at 1253, 1091, and 948 cm^−1^ resulted from the presence of C–O–C. There is no signal peak at 2270 cm^−1^ in all the spectra of the LPH film, which indicates that the –NCO groups completely reacted with the –OH group in PHA-diols or PEG polymer chains, and it converted into urethane linkage.

### 3.2. Thermal Properties and Crystallinity Analysis

TG and DTG analyses of LPH-0, LPH-2, LPH-4, LPH-6, and LPH-8 are illustrated in Figure 4A,B, respectively. The degradation of the LPH-0 occurred in only one step, while other LPH films appeared in three stages for thermal degradation. Table 2 shows the thermal properties parameters for LPH films. The T_onset_ (5%), T_d_ (20%), and T_dmax_ (80%) of PHA-based polyurethane were close to that of LPH-0. The LPH films with different PHA-diol content have similar thermal behavior under an N_2_ atmosphere. Three characteristic temperature regions were occurred at (1) 250–300, 340–360, and 380–430 °C, but the LPH-0 film has only one characteristic temperature regions at about 410 °C. Compared to LPH-0 sample, there is a clear difference on the first and second stage for the LPH films (from LPH-2 to LPH-8). The first step and second stage may correspond to the PHA-diol segment degradation in the structure of the samples. The third stage is similar to LPH-0 in regard to the degradation of the urethane linkages present in the structure.

The melting behavior and crystallization of LPH films are carried out by DSC and XRD. It is important that the crystallinity of the polymer materials affects the mechanical properties. The thermal melting behavior of the LPH films were analyzed by DSC. Figure 5A shows the second heating curves for LPH films. The corresponding melting parameters are also listed in Table 2. However, melting temperatures (Tm) are slightly different. The Tm of the LPH films decreased when the PHA-diol content increased, possibly because more cross-linking points in the LPH polymer back bone were created. It may limit the movement for the PEG polymer chain segment. In order to investigate the effect of the PHA-diol content about the crystalline ability for LPH films, their crystal structures were characterized by wide-angle X-ray diffraction(XRD). Figure 5B shows the XRD patterns of LPH films. The diffractograms show that LPH-0 sample are basically amorphous. There is only a broad diffraction band at 2θ values of 21.6° and no defined diffraction peaks occurred. However, LPH-2, LPH-4, LPH-6, and LPH-8 show that the crystalline zone exists in the polymer backbone. With PHA-diol content increasing, the intensity crystal of the LPH is better, which is consistent with the above DSC results.

### 3.3. Mechanical Properties of LPH Films

The mechanical properties is one of the most important factors in tissue engineering. The bone tissue engineering scaffold achieved the tensile strength requirement at the range of 8–20 MPa [42], and the Young’s modulus of samples at the range of 5–500 MPa [43]. The mechanical properties of LPH films were determined by quasi-static tensile. Typical stress–strain curves of LPH samples are shown in Figure 6A, and the measured mechanical properties are summarized in Table 3. The LPH-0 shows lower tensile Young’s modulus, tensile strength, and elongation at break. With the increasing of molar fraction of PHA-diol, the maximum tensile strength and Young’s modulus of the films increase gradually. The elongation at break of the samples first increases and then decreases. The maximum tensile strength and Young’s modulus for the LPH-8 are 9.49 and 25.61 MPa, much higher than that of other LPH films. The elongation at break of LPH-4 is 998%, but the LPH-8 is 424%. At the beginning, by increasing the content of PHA-diol, more chemical junction is formed in the chemical structure, so the elongation at break increased to some extent, but when the crystallinity reaches a certain extent, the sample begins to show the brittle behavior, so the elongation at break of the sample is decreased. These results indicated that the mechanical performance of the LPH films synergistically depended on the PHA-diol contents. Figure 6B shows the LPH-8 variation in the tensile storage modulus (E’) as a function of temperature. The modulus of the LPH film decreased with increasing temperature. The E’ of LPH-8 was 1.2 MPa at 25 °C. It indicates that the LPH-8 film is soft at room temperature. The curve of loss factor (tan δ) vs. temperature is consistent with the above DSC results. When the temperature exceeds 60 °C, the sample is in a molten state.

### 3.4. Swelling Properties of LPH Films

The hydrophilicity can influence the biocompatibility of the LPH materials. Water contact angle (*θ_H2O_*) has been measured to confirm the hydrophilicity or hydrophobicity of the LPH material. As is well-known, a *θ_H2O_* less than 90° is considered as the hydrophilic, and more than 90° as hydrophobic [44]. The water contact angles of LPH films samples were measured to characterize their surface hydrophilicity in Figure 7. It also showed a trend toward lower water-contact angles from the higher PHA-diol content in LPH polymer. LPH-2 sample had the highest water contact angle while LPH-0 had the lowest water contact angle. To verify the hydrophilicity changes with different PHA-diol content, we examined the water contact angles of the different LPH films. Before adding the PHA-diol into prepolymer, the LPH-0 sample was relatively hydrophilicity with a water contact angle of 47° ± 0.3°. Adding PHA-diol into the prepolymer could increase the hydrophobicity for LPH film. For example, the LPH-8 films caused the water contact angle to increase by about 30°compared to LPH-0 film. This changes confirmed and could explain that the PHA has a hydrophobic character [45], but with the increasing of PHA-diol content, the water contact angle will decrease. This may be due to the roughness of the material surface, based on the SEM results. The water contact angles for LPH-0 and LPH-8 films were in the reference range (45°–76°) with proper cells adhesion in biomedical applications [46].

The water absorption measurement of LPH films is very important because the mechanism for hydrolytic degradation of the polyurethane polymer is hydrolysis of ester and urethane groups. The water absorption of LPH films is carried out in order to evaluate the moisture-absorption capacity when it is in the pH 7.4 PBS solution at different temperatures (4, 25, and 37 °C). Water absorption for various LPH films in water of different temperatures is listed in Figure 8. We found that the LPH-0, LPH-2, LPH-4, LPH-6, and LPH-8 films separately absorbed water at pH 7.4 and 37 °C of 77%, 121%, 128%, 142%, and 151%. It is because many hydrophilic groups, such as hydroxyl and ester groups, existed in the LPH polymer backbone. In addition, adding the PHA-diol chemical composition partly improved the water absorption. The water absorption of LPH film increased by 74%, while the PHA-diol content increased from 0% to 14.5%. Above all, some water absorption value means the LPH polymer can be easily subjected to hydrolytic degradation. The swelling capacity revealed LPH-8 could reach the highest value (184%) at 4 °C. Low temperature is good for water absorption. This is due to crystallization of PEG, and there are a great number of hydroxyl functional groups in the polymer backbone segment [47].

### 3.5. In Vitro Degradation of LPH Films

In vitro degradation experiments of the LPH samples were carried out by submerging in 0.1 M phosphate buffer solution (PBS) at 37 °C. The morphology of the LPH films before and after 49 days of degradation was studied. Figure 9 shows that the surfaces of the LPH films were different before degradation. The surface of LPH-0 was wrinkled, and the other surface of LPH was flat, but there were many small dendritic chain segments penetrating the surface. By increasing the content of PHA-diol, the small dendritic chain segments of the surface increase, while the surface of LPH films becomes rougher and rougher. After 49 days degradation in PBS solution, it can be seen that the surfaces of the LPH samples showed clearly different morphological changes from Figure 9a–e. There were many short chain segments on the LPH-0 surface. This may be due to its low degree of cross-linking, but the other LPH samples showed short polymer chain frameworks and backbones on the erosion surface. It also can be seen that, as the PHA-diol content increases, the corrosion degree for LPH film increases. There are some holes on the LPH-8 surface; this may indicate that the LPH-8 degrades fastest.

Basing on the analyses above, it can be concluded that LPH-8 has the fastest degradation rate. In order to investigate the effect with different pH values for degradation, the morphology changes of the same LPH-8 film after degradation in different pH solution (pH =3, 7.4, and 11) was also studied. As seen in Figure 10, the LPH-8 film showed obvious different morphologies under different pH values of PBS solution, after 49 days of degradation. It can be seen that the surfaces corrosion morphology of the LPH-8 film has a significant difference. In Figure 10A, the surfaces of LPH-8 were partially dissolved in pH of 11 PBS solution, which might be in dendrite exfoliates form. From Figure 10B, it can be seen that a great number of deep grooves are on the surfaces in pH of 7.4 PBS solution. It might degrade by exfoliation in sheet form, but there were many pellets on the surface of LPH-8 in pH of 3 in Figure 10C. When the LPH sample was in pH of 3 solution for a long time, part of the material might fall off in pellet form. This suggests that they are prone to corrosion at different rates and compositions in different pH solutions.

### 3.6. Cytocompatibility of the LPH Films

The practical applications of LPH samples require good cytocompatibility when they are used as tissue scaffolds. Figure 11 shows the CCK-8 results in vitro cell viability of the LPH-0, LPH-2, LPH-4, LPH-6, and LPH-8 films for 1, 3, and 7 days, respectively. There is no obvious difference in cell viability between the different LPH samples on day 1, but the optical density (OD) value of cell proliferation for the LPH films with PHA-diol content increasing was slightly higher. Even some LPH samples were better than the positive control TCPS. The OD value of cell proliferation for all samples at 3 days is higher than the first day. At day 7, it can be seen that a clear trend about the cell proliferation for the LPH films increases, and LPH-8 show the best cell viability. Over a 7-day period, the optical density value of cell proliferation for the rBMSCs significantly increased for each of the LPH samples and the control group of TCPs. Therein, the OD value of theLPH-8 sample is consistently higher than that in the control group TCPs, at different days, indicating that LPH-8 shows the best biocompatibility.

The rBMSCs were cultured on the surface of LPH films, to evaluate the cytocompatibility. Figure 12 shows the morphologies of rBMSCs on pristine LPH-0 and LPH-8 films on 1, 3, and 7days. As can be seen from Figure 12, some sporadic cells with a spindle and spherical shape morphology are found attached on the surface of the LPH-0 and LPH-8 films at day 1. Compared to the LPH-0, more rBMSCs are attached to the LPH-8 film surface after 3 and 7 days of growth. In addition, the rBMSCs on LPH-8 have more extended filopodia and spread better. This indicates that the LPH-8 film is more likely to adhere to cells and facilitate cell growth.

## 4. Conclusions

Stretchable, biodegradable, and biocompatible polyhydroxyalkanoates-based polyurethane (LPH) was synthesized with L-lysine diisocyanate, poly(ethylene glycol), and polyhydroxyalkanoate. The physicochemical and mechanical aspects, swelling capacity, degradation, and cell viability of LPH copolymers were studied. This study demonstrated that the mechanical properties, hydrophilic performance, degradation behavior, and cytocompatibility properties of LPH films could be adjusted by tailoring the PHA-diol compositional gradient. With the increasing of the PHA-diol content, more crystalline domains formed. Furthermore, as the PHA-diol content increases, the Young’s modulus and the tensile strength of the LPH films improve. When more PHA-diol content was introduced into the LPH, the hydrophilic performance and the degree of degradation increased. The degradation of the LPH films was associated with their water-absorption properties. The degradation rate is also affected by different pH conditions. In conclusion, the LPH-8 film has excellent cytocompatibility, which could be used as tissue engineering scaffold.

## Figures and Tables

**Figure 1 polymers-11-01927-f001:**
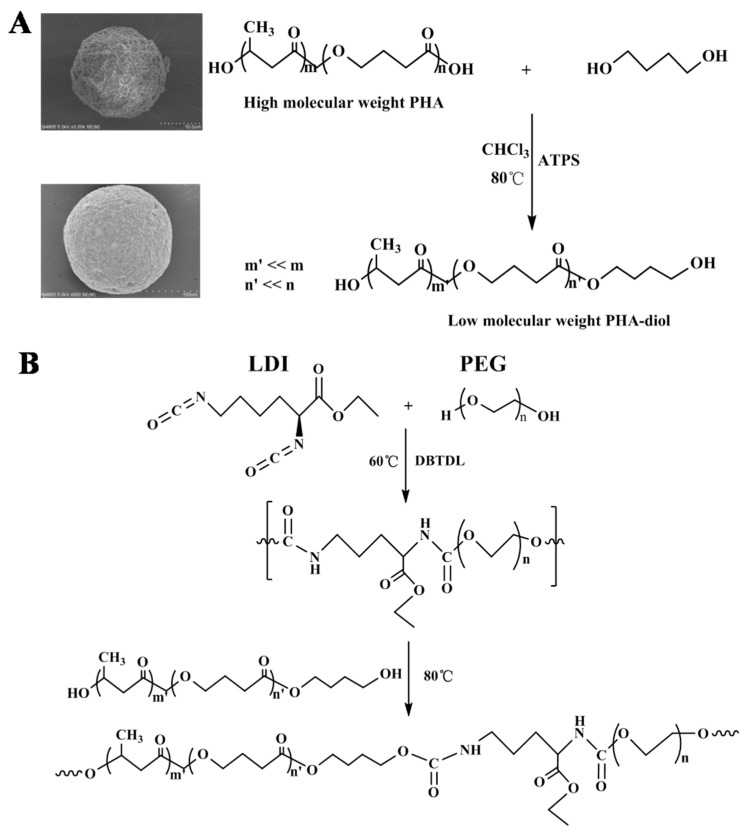
(**A**) Synthesis scheme of PHA-diol and (**B**) synthesis scheme of LPH.

**Figure 2 polymers-11-01927-f002:**
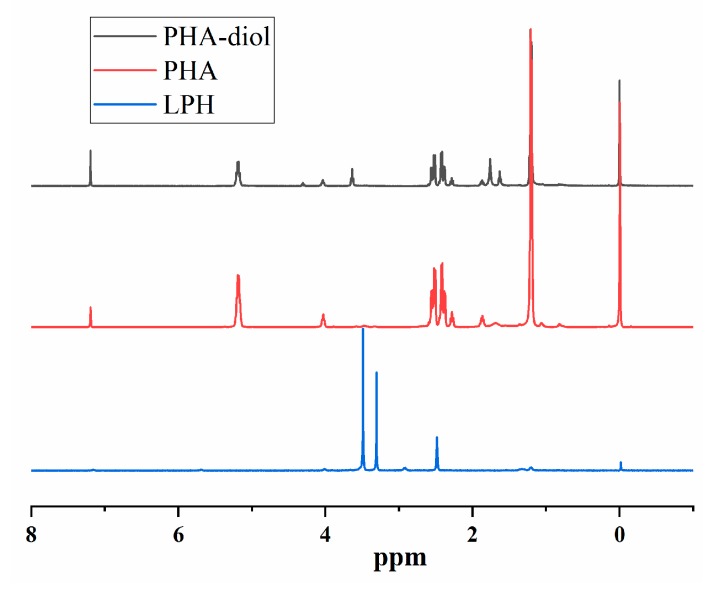
^1^H-NMR spectra of PHA, PHA-diol in CDCl, and LPH-8 in DMSO-d_6_.

**Figure 3 polymers-11-01927-f003:**
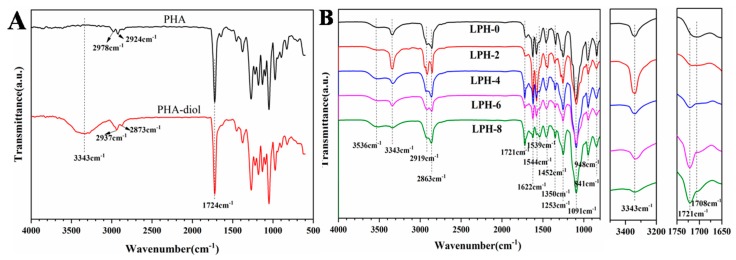
ATR-FTIR spectra of (**A**) PHA, PHA-diol, and (**B**) LPH films.

**Figure 4 polymers-11-01927-f004:**
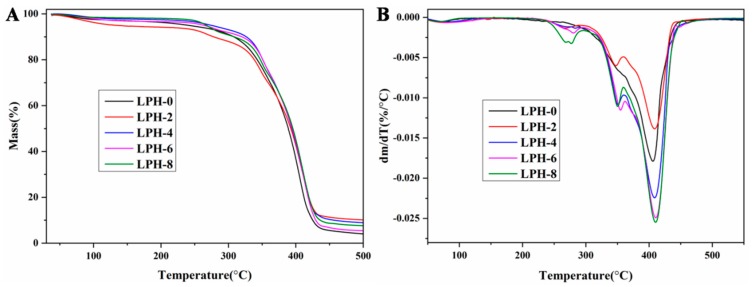
(**A**) TG and (**B**) DTG thermal property curves of the LPH films.

**Figure 5 polymers-11-01927-f005:**
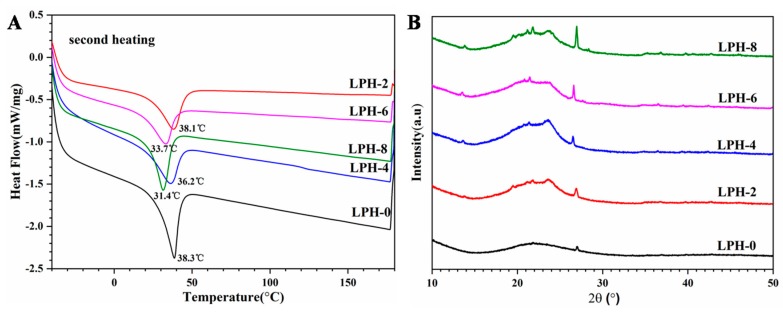
(**A**) The DSC and (**B**) XRD of the LPH films.

**Figure 6 polymers-11-01927-f006:**
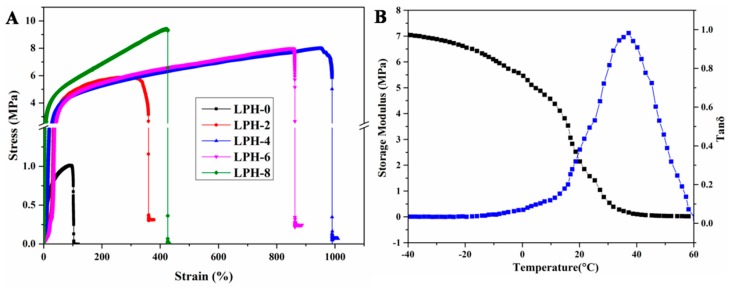
(**A**) The tensile stress–strain curves of LPH films and (**B**) dynamical mechanical analysis of LPH-8 for storage modulus and tan δ.

**Figure 7 polymers-11-01927-f007:**
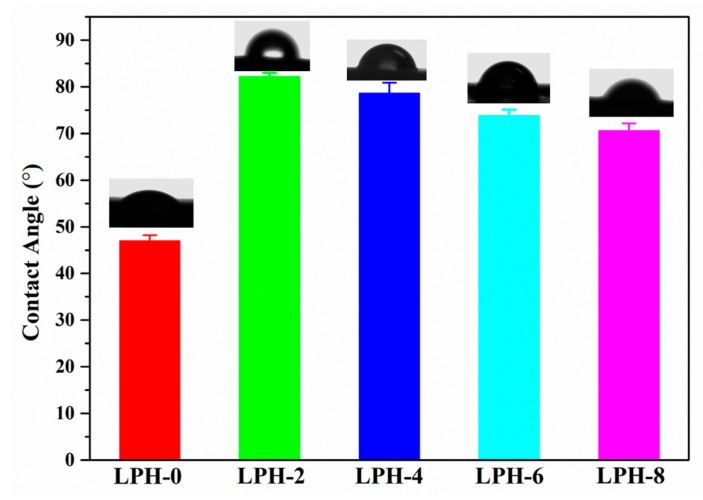
Representative images of water contact angle of LPH-0, LPH-2, LPH-4, LPH-6, and LPH-8 films.

**Figure 8 polymers-11-01927-f008:**
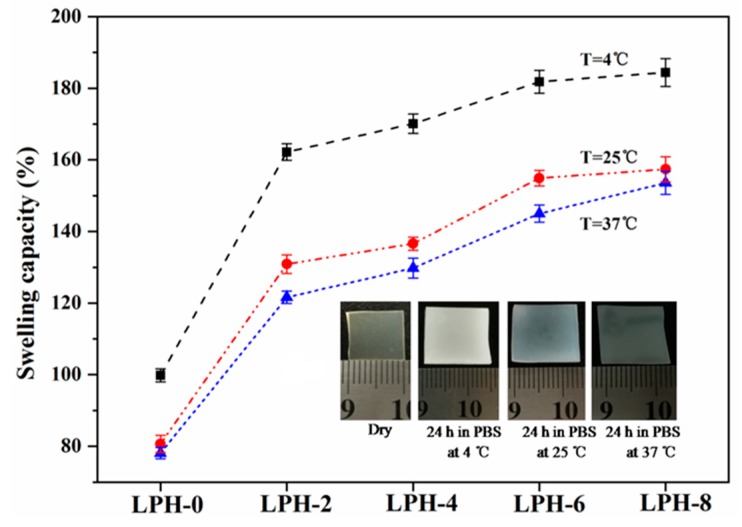
Water absorption of the LPH films at different temperatures.

**Figure 9 polymers-11-01927-f009:**
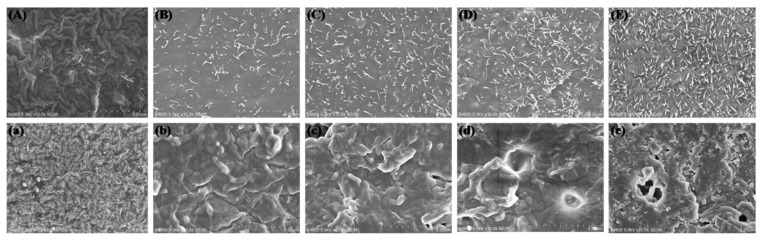
The SEM images of all LPH films’ surface in PBS solution (pH = 7.4) in vitro before and after degradation (**A**–**E** is from LPH-0 to LPH-8 before degradation; **a**–**e** is from LPH-0 to LPH-8 after 49 days degradation; all images magnified 10.0k).

**Figure 10 polymers-11-01927-f010:**
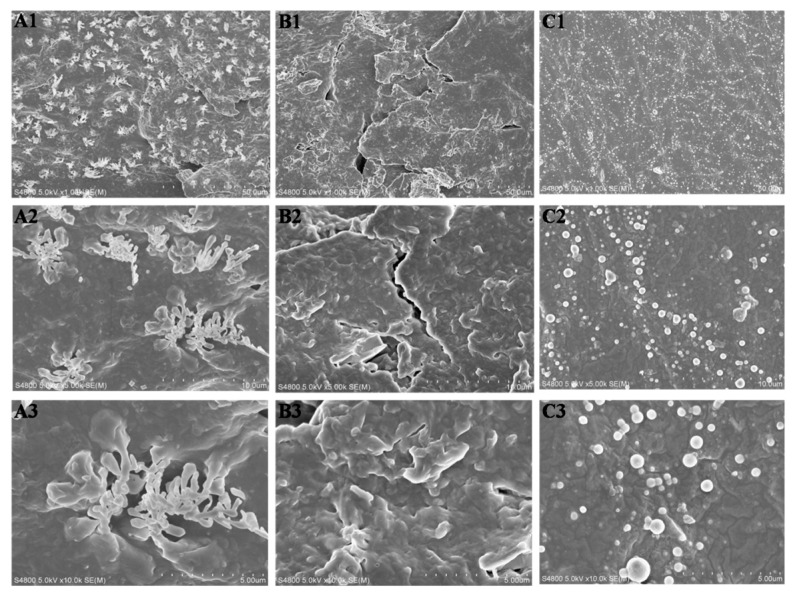
The SEM of the surface ofLPH-8 film after 49 days in vitro degradation (**A**) in pH of 11PBS solution; (**B**) in pH of 7.4 PBS solution; (**C**) in pH of 11 PBS solution. (**A1**,**B1**,**C1**) magnified 1.0k; (**A2**,**B2**,**C2**) magnified 5.0k; (**A3**,**B3**,**C3**) magnified 10.0k.

**Figure 11 polymers-11-01927-f011:**
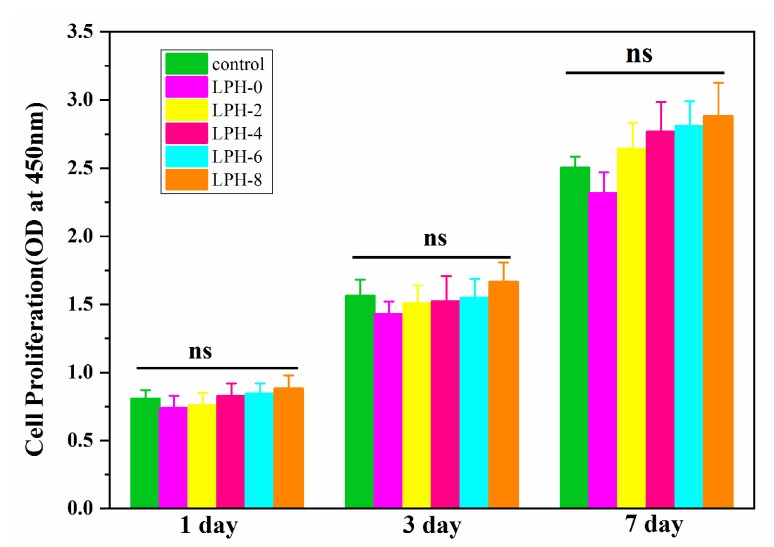
The viability of rBMSCs cultured on the LPH scaffolds after different cell-culture periods.

**Figure 12 polymers-11-01927-f012:**
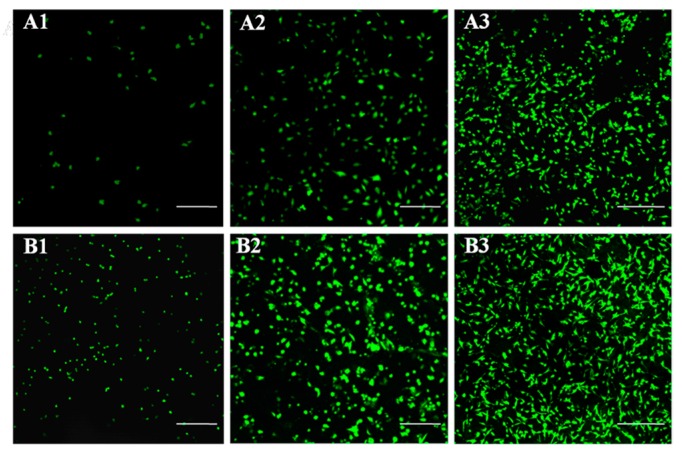
Cytocompatibility of rBMSCs on LPH-0 and LPH-8 samples for different days. The scale bars: 300 μm. (**A1**–**A3**) LPH-0 cultured at 1, 3, and 7 days; (**B1**–**B3**) LPH-8 cultured at 1, 3, and 7 days.

**Table 1 polymers-11-01927-t001:** The formulation of synthesized LPH films.

Samples	Feed Ratio (mmol)				Molecular Weights		
	LDI	PEG	PHA-diol	NCO/OH	Mw (g mol^−1^)	Mn (g mol^−1^)	Mw/Mn
LPH-0	2	1	0	2	25,800	24,500	1.05
LPH-2	2	1	0.2	1.67	53,600	42,200	1.27
LPH-4	2	1	0.4	1.43	125,000	116,000	1.08
LPH-6	2	1	0.6	1.25	176,000	137,000	1.28
LPH-8	2	1	0.8	1.11	202,000	152,000	1.33

**Table 2 polymers-11-01927-t002:** Thermal parameters of LPH films by TGA and DSC measurements.

Sample	T_5%_ (°C)	T_20%_ (°C)	T_80%_ (°C)	Tm (°C)	∆Hm (J/g)
LPH-0	241	342	412	38.3	55.93
LPH-2	134	338	419	38.1	50.15
LPH-4	275	351	419	36.2	45.32
LPH-6	264	351	418	33.7	39.15
LPH-8	268	347	419	31.4	46.61

**Table 3 polymers-11-01927-t003:** Mechanical properties of PHA-based polyurethane films.

Sample Code	Young’s Modulus (MPa)	Tensile Strength (MPa)	Elongation at Break (%)
LPH-0	3.07 ± 0.4	1.01 ± 0.06	102.79 ± 6.4
LPH-2	9.01 ± 1.1	5.67 ± 0.08	338.11 ± 16.2
LPH-4	10.65 ± 1.8	7.86 ± 0.18	998.64 ± 32.3
LPH-6	11.21 ± 1.6	8.07 ± 0.14	863.01 ± 19.9
LPH-8	25.61 ± 2.7	9.49 ± 0.17	424.01 ± 28.7

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
