# Peer review of "Development of Nontoxic Biodegradable Polyurethanes Based on Polyhydroxyalkanoate and L-lysine Diisocyanate with Improved Mechanical Properties as New Elastomers Scaffolds"

_polymers, 2019, doi:10.3390/polym11121927_

Round 1

Reviewer 1 Report

The manuscript entitled “Development of nontoxic biodegradable polyurethanes based on polyhydroxyalkanoate and L-lysine diisocyanate with improved mechanical properties as new elastomers scaffolds” presents a new type of polyurethanes based on polyhydroxyalcanoates very versatile and useful in the biomedical sector. In general, the present manuscript is well-written and structured, allowing easy reading and interpretation of the results.

The present manuscript should be published after the authors take into account the following remarks.

Table 1 shows the moles of the reagents used. It is surprising that a stoichiometric relationship between isocyanate groups and alcohol groups is not maintained. This, in principle, should lead to a stoichiometric imbalance that would result in a polyurethane with low performance and mechanical properties. However, the mechanical properties of the materials are very good. Therefore, authors should review the values in Table 1. In Figure 2, the authors show the spectrum of the LPH polyurethane and describe the positions of each component. However, in that LPH spectrum only 3 peaks are observed, therefore the authors should change that spectrum to another one, with better quality, to maintain the text, in which they describe the polyurethane. In Figure 3, the authors describe the positions of the infrared bands of the prepared polyurethanes. The band at 1721 cm-1 is associated, according to the authors, with the free carbonyl groups, however, the authors have not taken into account that this band is also present with the carbonyl bonds of the polyester. Therefore, this band increases in intensity as the polyhydroxyalkanoate content increases in the final polyurethane. Therefore, please correct this observation. This reviewer has noticed the characterization of the polyhydroxyalkanoate-diol is missing. The content of OH groups, their average molecular weight in number or their final composition does not appear. This is very important to then calculate the values shown in Table 1. Therefore, the authors should include the information corresponding to polyester diol, OH value and Mn. Finally, the authors have shown in table 3, the values of the mechanical properties of the materials as a function of the polyhydroxyalkanoate content. It should be noted that as the polyester content increases, the mechanical properties improve. However, I would like to know if the authors have other polyurethanes with higher soft segment content, that is, I would like to know where the maximum of these materials is.

Author Response

Response: Thanks for the reviewer’s question and advice. We added the value of NCO/OH and Mn. We have revised the Table1 as follow(see page 4, line 99). In Figure 2 of LPH spectrum, not only three peaks are observed. There are other faint peaks that we refer to in the manuscript. In Figure 3,We have added the ATR-FTIRspectra of PHA and PHA-diol and some new discussions in the revised manuscript(see page 6, line 181-186) as follows. " ATR/FT-IR was also used to characterize the structure of the synthesized PHA-based polyurethanes. In Fig.3A, the spectra of PHA and PHA-diol is presented. There is one obvious difference for PHA and PHA-diol at 3343 cm−1. The broad peak is the characteristic adsorption of  -OH stretching vibrations in the spetra of PHA-diol. For PHA, there is no peak at about 3343 cm−1. It means there is no hydroxyl group in PHA. A strong and sharp absorption band at 1724 cm−1 in the spetra of PHA and PHA-diol is observed in Figure 3A, which is the-C=O stretching peaks. " In table 3,the polyester content increases, the Young’s modulus  and the tensile strength improve. But the elongation at break decline. So it's not that the higher the polyester content, the better the mechanical properties.

Author Response

Point 1: What does “LPH” mean? The letter “L”, “P”, and ”H” are the abbreviation for what? I guess the L stands for L-lysine diisocyannate and the P means poly (ethylene glycol), but what is H? Moreover, the abbreviation should appear after the full name and not before it.

 Response: Thanks for the reviewer’s advice. Yes, L stands for L-lysine diisocyannate, the P means poly (ethylene glycol) and H stands for polyhydroxyalkanates. The abbreviation LPH is first appeared in the abstract and has been given its full name.

Point 2: Please add the parameters used in the method of SEM.

 Response: Thanks for the reviewer’s suggestion. We have added it in the revised manuscript (see page4, line118-121) as follows: “Morphologies of the LPH films before and after degradation were observed by scanning electron microscope (SEM, Hitachi S-4800, Japan) at an accelerating voltage of 5.0 kV. The LPH films were prepared by  drying in 60 ℃ oven for 24h and the  surfaces of the samples before and after degradation by sputtering gold for 50s.”

Point 3: The scale bars are not clear in SEM images and the images for cells. If authors think the scale bar does not look good in the image, please add the information in the caption.

Response: Thanks for the reviewer’s advice. We have added it in the revised manuscript (see page 11, line315-317). ”Figure 9. The SEM images of all LPH films'surface in PBS solution( pH=7.4)in vitro before and after degradation (A, B, C, D, E is from LPH-0 to LPH-8 before degradation; a, b, c, d, e is from LPH-0 to LPH-8 after 49 days degradation; all images magnified 10.k).”

Point 4:What is the size for the samples used for the mechanical test? Then, the cross-head speed can be converted to strain rate. After this, the cross-head speed make more sense. 

Response: Thanks for the reviewer’s suggestion. We have modified it in the revised manuscript (see page 4, line122-126). ”Tensile testing was performed with CMT4204 (China) at room temperature. The LPH samples were obtained by the dumbbell-shaped mold (50mm×8.5mm×2mm; narrow parallel section 16mm×4mm) after solvent evaporation. The tensile properties of the LPH films were evaluated by a SANS system at room temperature, equipped with a 2 kN load force. The strain rate was 0.01 s-1. The tension test for each LPH sample was repeated three times.”

Point 5:How did the authors stain the cell? The images look like fluorescence or confocal images. Please confirm this in the method. What emission light is used? I suppose the cells were stained first. If yes, please add the process of staining in the method.

Response: Thanks for the reviewer’s suggestion. We have revised it in the revised manuscript (see page 5, line 155-159). “Cultivation of the rBMSCs was designed to study the possible effects of LPH samples in the cell proliferation experiments. rBMSCs were first stained with Celltracker-green. Then the rBMSCs were cultured on the surface of the LPH samples in a density of 1×105 cells/ml. The morphology of the rBMSCs was observed by confocal laser scanning microscope (CLSM)(Leica, TCS SP5â…¡, Germany) on 1, 3 and 7 days respectively.

Point 6: Authors used the casting film for all of tests. It may not good for the mechanical test and the contact angel. Since the new polymer is a plastic, I suggest making the hot press film for these two characterizations. Especially, for the contact angel, authors already mention the results may due to the roughness. The good samples should be obtained before the discussion of the materials.

Response: Thanks for the reviewer’s advice. According to the reviewer 's suggestion, the hot press film may be better for the mechanical test and the contact angel.The author totally agree with it. In the following study, the author will compare the effects of the the mechanical test and the contact angel for the casting film and the hot press film. However,it is rational for using the casting film for the mechanical test and the contact angel. Many published papers can support it (10.1023/b:jooe.0000003121.12800.4e; https://doi.org/10.1007/s10853-019-03521-9;  DOI: 10.1039/c6py01131d).

Point 7:The mechanical property of the new polyurethanes is enhanced. As a structural material, stronger and tougher is better property. However, as a biological scaffold, stronger material may be good but the stiffness of material effects the differentiation, proliferation and attachment of the stem cells. What strength, modulus and elongation are needed for the scaffold or regenerative medicine need to be introduced first.

 Response: Thanks for the reviewer’s advice. We have added it in the revised manuscript (see page 8, line242-244). "Mechanical properties is one of the most important factors in tissue engineering. The bone tissue engineering scaffold achieved the tensile strength requirement at the range of 8-20 MPa [42]and the Young’s modulus of samples at the range of 5-500 MPa[43]."
